# LEARNING SUBGOAL REPRESENTATIONS FROM STATE GRAPHS IN GOAL-CONDITIONED HIERARCHICAL REINFORCEMENT LEARNING

## ABSTRACT

The integration of graphs with Goal-conditioned Hierarchical Reinforcement Learning (GCHRL) has recently gained attention, as the intermediate goals (subgoals) can be effectively sampled from graphs that naturally represent the overall task structure in most RL tasks. However, some existing approaches often rely on domain-specific knowledge to construct these graphs, limiting their applicability to new tasks. Other graph-based approaches create graphs dynamically during exploration but struggle to fully utilize them because they have problems passing the information in the graphs to newly visited states. Additionally, current GCHRL methods face challenges such as sample inefficiency and poor subgoal representations. In this paper, we present a solution to these issues through the development of a graph encoder-decoder that can evaluate unseen states. Our proposed method, Graph-Guided sub-Goal representation Generation RL (G4RL), can be incorporated into any existing GCHRL method to enhance performance. We show that the graph encoder-decoder can be effectively implemented using a network trained on the state graph generated during exploration. Empirical results indicate that leveraging high and low-level intrinsic rewards from the graph encoder-decoder significantly enhances the performance of state-of-the-art GCHRL approaches in both dense and sparse reward environments.

## 1 INTRODUCTION

Traditional reinforcement learning methods face great challenges when learning policies in environments with long time horizons and sparse rewards. To address these challenges, Hierarchical Reinforcement Learning (HRL) methods have been proposed to break problems into smaller, more manageable scales suitable for efficient learning. Previous works (Sutton et al., 1999; Kulkarni et al., 2016; Vezhnevets et al., 2017; Levy et al., 2017) have shown its capability in large, sparse-reward environments. Among the HRL methods, Goal-Conditioned Hierarchical Reinforcement Learning (GCHRL) has attracted much attention because of its well-defined paradigm (Nachum et al., 2018b) and resemblance to the human thinking process. Although GCHRL has shown superior performance compared to non-hierarchical methods in certain scenarios, questions such as how to find better subgoal representations and explore the state space more efficiently (Nachum et al., 2018a; Guo et al., 2021) remain unanswered.

GCHRL methods typically utilize two levels of agents as stated in Nachum et al. (2018b). A high-level agent chooses the next target state based its current state, while a low-level agent decides how to reach the target state specified by its high-level instructor. Both agents face their own challenges. The high-level agent suffers from sample inefficiency in environments with a large state-action space, a problem also encountered by non-hierarchical methods, while the low-level agent is trained solely by the reward signals derived from the distances between the current state and subgoals in the representation space, making the reward signals highly susceptible to bad subgoal representations.

Previous works have tried to either enhance the efficiency of high-level exploration (Huang et al., 2019; Zhang et al., 2022) or find a good subgoal space to boost GCHRL's performance in complex environments (Wang et al., 2024). To the best of our knowledge, while these methods have made

progress in their respective areas, no prior work has attempted to address them in combination. This integration is crucial, as uniting an effective high-level agent with an accurate low-level agent within a cohesive framework can yield performance improvements that surpass simple additive effects.

Combining graph theory with RL has recently become a trend in the community (Lee et al., 2022; Gieselmann & Pokorny, 2021) as graphs are inherently well-suited for representing the environment and task structure.

Previous work has focused on areas such as decision-making through graph search or traversal (Wan et al., 2021; Shang et al., 2019; Eysenbach et al., 2019), and as well as using graphs as world models (Zhang et al., 2021; Huang et al., 2019).

Yet, many previous works rely on pre-crafted graphs which hinders the generalization of the methods. Additionally, most existing works use graphs constructed directly from the original state space, which cannot provide meaningful guidance when the agent encounters a state that is not represented in those graphs.

To utilize graph representations when a new state (node) is encountered, switching to these representations (Hamilton, 2020; Chen et al., 2020; Khoshraftar & An, 2024) through graph learning is a viable option. Recent studies have shown that employing various types of graph representations for learning can benefit the underlying RL problem (Klissarov & Precup, 2020; Klissarov & Machado, 2023).

In this paper, we propose a novel approach that simultaneously addresses all of the aforementioned problems. Specifically, we keep track of visited states, construct a state graph based on them until the number of nodes reaches a threshold, and then update the graph by adding and dropping nodes as new states are visited, all while building a subgoal space through graph learning based on state representation and spatial connection information provided by the graph. By generating subgoal representations through graph learning, these representations will reflect their relative positions in the decision chain, thereby creating a more effective subgoal space. In estimating the distance between the current state and the intended subgoal in the original space, we use the distance between their corresponding representations in the subgoal space. This distance is then used to calculate the intrinsic reward for curiosity-driven exploration, aiming to improve sample efficiency.

The main contributions of this paper are as follows:

- We propose a novel architecture that employs a graph encoder-decoder to summarize spatial information into subgoal representations, enabling the evaluation of newly visited states. This architecture can be integrated into any existing GCHRL algorithm to enhance performance.
- We present a method for online construction of the state graph as a world model (Ha & Schmidhuber, 2018; Zhang et al., 2021) for the agent by sampling from trajectories.
- We use novelty-based auxiliary rewards (Simsek & Barto, 2006; Nehmzow et al., 2013) derived from subgoal representations to improve sample efficiency for high-level and low-level agents.

We tested our approach on several real-world application environments (Todorov et al., 2012) to assess the significance of our experimental results. The findings indicate that our method can significantly enhance the performance of the underlying HRL approach in terms of both sample efficiency and success rate/cumulative reward.

## 2 PRELIMINARIES

### 2.1 NOTATION

As the most common way to model reinforcement learning scenarios, we introduce the Markov Decision Process (MDP) (Puterman, 2014) defined as a tuple $< \mathcal{S}, \mathcal{A}, P, R, \gamma >$ as follows: At each time step $t$, the agent observes the current state $s_t \in \mathcal{S}$ provided by the environment and chooses an action $a_t \in \mathcal{A}$ based on its internal policy $\pi(a_t|s_t)$, which defines the probability of selecting an action given the current state. The action is then executed, and the interaction with the environment leads the agent to a new state $s_{t+1}$ according to a transition probability function

$P(s_{t+1}|s_t, a_t)$ which is known only to the environment. Subsequently, the agent receives a reward $r_t$ from the environment determined by a reward function $R(s_t, a_t)$ that evaluates the action taken in the current state and is also only visible to the environment. The agent aims to learn an optimal policy $\pi$ to maximize the expected discounted cumulative reward $\mathbb{E}_{\pi, t:0 \to T}\left[\sum_{i=0}^{T} \gamma^i r_i\right]$, where $\gamma$ (with $0 \le \gamma < 1$) is a pre-defined discount factor used to prioritize immediate rewards over distant future rewards, therefore preventing the total reward from becoming infinite.

## 2.2 GOAL-CONDITIONED HIERARCHICAL RL (GCHRL)

Goal-Conditioned Reinforcement Learning (GCRL) trains agents to achieve specific goals, which are the target states. The agent receives an additional goal input along with the state input and learns a policy to achieve this goal. Goals are represented explicitly in the input to the policy, guiding the agent's actions towards achieving specific outcomes, and the reward function is often goal-dependent, providing positive feedback when the agent reaches the desired goal state.

To deal with large and complicated environments, Goal-Conditioned Hierarchical Reinforcement Learning (GCHRL) (Nachum et al., 2018b; Zhang et al., 2022; Wang et al., 2024) decomposes the learning task into a hierarchy of smaller, more manageable sub-tasks. Typically there are two levels of agents. At time step $t$, the high-level agent chooses a subgoal $g_t$ which is the state representation of a target state that the high-level agent wants the low-level agent to achieve as a part of the overall task. This choice is made by sampling $g_t$ from the high-level policy $\pi_h(g_t|\phi(s_t))$, where $\phi : s \mapsto \mathbb{R}^d$ is the state representation function which gives a condensed representation of the state.

Each state $s_t$ can be mapped to its subgoal feature $g(s_t)$ by a subgoal feature extractor. Note that $g(s_t)$ is not the same as $g_t$. The former, $g(s_t)$, is the learned subgoal feature of the current state $s_t$, while the latter, $g_t$, is the target state we aim to reach from the state $s_t$ in one step or multiple steps.

Given the subgoal $g_t$ sampled from the high-level policy $\pi_h(g_t|\phi(s_t))$ for the current time step $t$ and the state representation vector $\phi(s_t)$, a low-level agent executes action $a_t$ based on the low-level policy $\pi_l(a_t|\phi(s_t), g_t)$. The low-level agent is trained using the intrinsic reward signal $r_{int}(s_t, g_t, a_t, s_{t+1}) = -\|\phi(s_{t+1}) - g_t\|^2$ to encourage it to achieve the subgoal.

Both agents can be implemented by any policy-based methods, including those introduced in previous works on policy gradients such as Fujimoto et al. (2018); Haarnoja et al. (2018) and Schulman et al. (2017).

## 2.3 GRAPH AND MDP

Graph is a generic data structure, which can model complex relations among objects in many real-world problems. A graph is defined as $\mathcal{G} = (\mathcal{V}, \mathcal{E})$, where $\mathcal{V} = \{1, 2, \dots, N\}$ is the set of nodes and $\mathcal{E} = \{e_{ij}\}$ is the set of edges without self-loops. The adjacency matrix of $\mathcal{G}$ is denoted by $\boldsymbol{A} = (\boldsymbol{A}_{i,j}) \in \mathbb{R}^{N \times N}$ with $\boldsymbol{A}_{i,j} = 1$ if there is an edge between nodes $i$ and $j$, otherwise $\boldsymbol{A}_{i,j} = 0$. The adjacency matrix can be extended to a *weighted* adjacency matrix, where $\boldsymbol{A}_{i,j}$ is a weight of the edge $e_{ij}$.

In MDP, a node can represent a state, while the edge weights can model the transition probabilities or reachability statistics between states.

## 3 METHODS

This section presents our framework, Graph-Guided subGoal representation Generation (G4RL). Our method reshapes the subgoal space utilizing a state graph to incorporate the relative spatial information of visited states.

One drawback of previous hierarchical reinforcement learning algorithms (Nachum et al., 2018b; Kim et al., 2021; Zhang et al., 2022; Luo et al., 2024) is the Euclidean distance calculated in the original state representation space between the current state and the intended goal does not reflect the true progress of the low-level agent, as there is rarely a straight path in the space from the current state to the subgoal. A low-level agent trained with such information may receive an inaccurate reward signal, thus impairing its performance. Another issue is that the high-level agent may propose

a subgoal that is too difficult to reach if its output is not constrained, wasting exploration steps on trying to reach those unreasonable subgoals (Zhang et al., 2022). Our proposed method aims to mitigate both problems by calculating the distance in a subgoal representation space between subgoal representations given by a graph encoder-decoder. This graph encoder-decoder takes into account the actual connectivity between states, ensuring that the generated subgoal representations respect adjacency information. This approach provides more accurate reward signals and leads to more reasonable target subgoals.

## 3.1 STATE GRAPH

To record the visited states and their connections, we maintain a state graph $\mathcal{G} = (\mathcal{V}, \mathcal{E})$ with a fixed number $N$ of nodes[1]. This graph is built and updated during training, with no pre-training using expert data or handcrafted process involved in its construction.

Each node is labelled by the corresponding state and for each node $s_t$, the corresponding state representation vector $\phi(s_t)$, which is also referred to as the node feature, is stored. Edges in the graph represent connectivity between states. The graph is constantly updated during exploration.

### 3.1.1 GRAPH CONSTRUCTION

The graph is initialized with $N$ empty nodes and no edges. The corresponding weighted adjacency matrix $\boldsymbol{A}$ is set to an $N \times N$ zero matrix. We perform the GCHRL explore process using randomly initialized policy $\pi_h$ and $\pi_l$. Once the agent encounters a state representation never seen before, that is, the representation vector is different from any state representations stored in the graph, as described in equation (1), it stores the state representation $\phi(s_t)$ as the node feature of an empty node in the graph and build an edge between this node and the node corresponds to the previous state.

$$\forall_{s_v \in \mathcal{V}}, \|\phi(s_t) - \phi(s_v)\|_2 > \epsilon_d, \tag{1}$$

$$\boldsymbol{A}_{s_t, s_{t-1}} = \boldsymbol{A}_{s_{t-1}, s_t} = 1, \tag{2}$$

where $\epsilon_d$ is a hyper-parameter controlling the distance threshold between state representations. When the agent encounters a state $s_t$ with feature $\phi(s_t)$ that is similar to several node representations already stored in the graph, it finds the state whose representation is the closest to the current state feature:

$$s_v = \underset{s_u : \|\phi(s_t) - \phi(s_u)\|_2 \leq \epsilon_d}{\arg\min} \|\phi(s_t) - \phi(s_u)\|_2. \tag{3}$$

Then the node $s_v$ is relabeled as $s_t$ and the weight for the edge $(s_{t-1}, s_t)$ is updated as follows:

$$\boldsymbol{A}_{s_{t-1}, s_t} = \boldsymbol{A}_{s_t, s_{t-1}} := \boldsymbol{A}_{s_{t-1}, s_t} + 1. \tag{4}$$

Note that a large weight indicates more frequent transitions between the underlying states.

We have used the Euclidean norm to define the distance between feature vectors. Since some elements may contain more spatial information than others, one can use a weighted Euclidean norm to define the distance between state representations instead.

### 3.1.2 GRAPH UPDATING

The graph has a fixed number of nodes. Suppose the graph is now full. When a new state $s_t$ is encountered, if $s_v$ from equation (3) exists, as before we relabel the node as $s_t$ and perform edge update as shown in equation (4); Otherwise, we replace the oldest state node in the graph with the current state node, delete all edges previously linked to that node, and create an edge $(s_{t-1}, s_t)$ with weight $\mathbf{A}_{s_{t-1}, s_t} = \mathbf{A}_{s_t, s_{t-1}} = 1$. Alternatively, we could replace the state node most weakly connected to other nodes by finding the node with the least weighted sum of edges.

---

[1]The number of training states for the graph encoder-decoder grows quadratically with $N$ because the adjacency weight matrix has $N^2$ elements. The choice of $N$ depends on the machine's capabilities.

## 3.2 GRAPH ENCODER-DECODER

To enable the assignment of suitable subgoal representations to every possible state, including unseen ones, we use node representations and edges to train a graph encoder-decoder. The parameter updates of the graph encoder-decoder and the policies during policy training are performed alternately in each episode. We do not train the graph encoder using expert policies then freeze its parameters for policy training.

The encoder-decoder starts training after the graph is full and continues periodically after processing a few trajectories. Section 3.3 will show the details of the training schedule.

The encoder $\mathbf{E}$ maps every state representation $\phi(s)$ to a subgoal representation $g(s)$. We use a feed-forward network (FFN) with several layers as the encoder $\mathbf{E}$:

$$g(s) = \mathbf{E}(\phi(s)) = \text{FFN}(\phi(s)). \tag{5}$$

The weight parameters of the feed-forward network will be learned through training. The decoder $\mathbf{D}$ takes two subgoal representations as input and outputs the inner product of these two representations:

$$\mathbf{D}(g(s_u), g(s_v)) = g(s_u)^T g(s_v). \tag{6}$$

The aim is to use the encoder-decoder structure to predict node relations. Naturally we can use $\boldsymbol{A}_{s_u, s_v}$ as a measure of the relation between nodes $s_u$ and $s_v$. But for the sake of numerical stability in the training process, we use $\boldsymbol{A}_{s_u, s_v} / \max_{s_i, s_j} \boldsymbol{A}_{s_i, s_i}$ as a measure. Thus the loss function is defined as:

$$\mathcal{L} = \sum_{s_u, s_v \in \mathcal{V}} \left[ \mathbf{D}(g(s_u), g(s_v)) - \boldsymbol{A}_{s_u, s_v} / \max_{s_i, s_j} \mathbf{A}_{s_i, s_i} \right]^2. \tag{7}$$

This loss function can enforce the subgoal representation provided by the encoder to respect neighbouring features in the graph.

## 3.3 ADAPTIVE TRAINING SCHEDULE OF THE GRAPH ENCODER-DECODER

The data stored in the graph, which comprises state representations as node features and connection information in the weighted adjacency matrix $\boldsymbol{A}$, is constantly updated. Due to the nature of online training, the update rate varies across episodes. If we train the graph encoder-decoder at regular intervals, this may lead to high variance and data loss in earlier episodes, as well as over-training in later episodes. To avoid this, we propose an adaptive training schedule for the graph encoder-decoder in the following paragraph.

There are two types of data changing in the graph: edge update, as shown in equation (4), and node replacement. We introduce a variable $c$ to track the weighted number of data changes. Since the replacement of nodes has a much higher impact on the data than the edge update, we add $N - 1$ to $c$ if a node replacement occurs, and add $1$ to $c$ if an edge update happens:

$$c = \begin{cases} c + N - 1, & \text{if a node replacement happens,} \\ c + 1, & \text{if an edge update happens.} \end{cases} \tag{8}$$

When this variable exceeds a certain value, specifically a tolerance $\beta$ multiplied by the total number of non-diagonal elements $N^2 - N$ in the matrix $\boldsymbol{A}$. we perform one gradient update for the graph encoder-decoder, and then we reset $c$ to $0$.

## 3.4 HIERARCHICAL AGENT WITH GRAPH ENCODER-DECODER

Our proposed method involves traditional goal-conditioned settings and a subgoal representation extractor implemented by a graph encoder-decoder.

The high-level policy $\pi_h(g_t | \phi(s_t))$ nominates a subgoal every $K$ steps and is trained using environmental reward $r_{ext}$. The policy can be implemented by any policy-based RL algorithm that takes transition tuples $(s_t, g_t, a_t, r_t, s_{t+1}, g_{t+1})$ as input. To encourage it to propose a subgoal that is not too difficult to reach from the current state $s_t$ for more efficient exploration, we add an intrinsic term to the high-level reward, considering the distance between the subgoal features of $s_t$ and $g_t$ in the subgoal space:

$$r_h(s_t, g_t, s_{t+1}) = r_{ext} + r_{int} = r_{ext} + \alpha_h \cdot \mathbf{D}(\mathbf{E}(\phi(s_t)), \mathbf{E}(g_t)), \tag{9}$$

where $\alpha_h$ is a hyperparameter that controls the significance of the intrinsic term in the high-level reward.

The low-level agent $\pi_l(a_t|\phi(s_t), g_t)$, however, operates in the subgoal space. While it still takes $\phi(s_t)$ and $g_t$ as input and outputs an atomic action $a_t$, we compute the reward based on distances in the subgoal space:

$$r_l(s_t, g_t, a_t, s_{t+1}) = -\|\phi(s_{t+1}) - g_t\|^2 + \alpha_l \cdot \mathbf{D}(\mathbf{E}(\phi(s_{t+1})), \mathbf{E}(g_t)), \tag{10}$$

where $\alpha_l$ is a hyperparameter controlling the significance of the reward term in the low-level reward. By computing the intrinsic reward in the subgoal space rather than in the state space, the function provides high values when proposed subgoals are easy to reach from the current location and low values when subgoals are close in the original state space but difficult to reach from the current location. The low-level agent can also be any policy-based algorithm.

### 3.5 Algorithm: G4RL

Now we describe our proposed method in Algorithm 1.

---

**Algorithm 1** G4RL

---

**Require:** High-level policy $\pi_h(g|\phi(s))$, low-level policy $\pi_l(a|\phi(s), g)$, replay buffer $\mathcal{B}$,
      graph encoder $\mathbf{E}$, graph decoder $\mathbf{D}$, high-level action frequency $K$,
      significance hyperparameter $\alpha_h$ and $\alpha_l$, tolerance hyperparameter $\beta$,
      number of episodes $N$, number of steps in one episode $T$.

1:   $n = 0$
2: **while** $n \leq N$ **do**
3:      $t = 0$
4:      $c = 0$
5:      **while** $t \leq T$ **do**
6:          **if** $t \mod K = 0$ **then**
7:              Execute the high-level policy $\pi_h(g_t|\phi(s_t))$ to sample the subgoal $g_t$.
8:          **else**
9:              Keep the subgoal $g_t$ unchanged.
10:        Execute the low-level policy $\pi_l(a_t|\phi(s_t), g_t)$ to sample the atomic action $a_t$.
11:        Sample reward $r_t$ and next state $s_{t+1}$.
12:        Calculate $r_h(s_t, g_t, s_{t+1})$ and $r_l(s_t, g_t, a_t, s_{t+1})$ using (9) and (10).
13:        Collect experience $(s_t, g_t, a_t, r_h, r_l, s_{t+1})$ and update the replay buffer $\mathcal{B}$.
14:        Update node representations and edge weights using collected experience.
15:        Update $c$ using (8).
16:        **if** $c \geq \beta$ **then**
17:           Update graph encoder $\mathbf{E}$ with node representation and edge information in the graph.
18:           $c = 0$.
19:        $t = t + 1$
20:      Update low-level policy $\pi_l(a|\phi(s), g)$ using the chosen HRL algorithm.
21:      Update high-level policy $\pi_h(g|\phi(s))$ using the chosen HRL algorithm.
22:      $n = n + 1$

---

## 4 Experiments

In this section, we present our experiments that combine G4RL with existing GCHRL methods to demonstrate its positive effects on enhancing their performances. We also show that the generated graph reasonably represents the overall structure of the task. Our experiments are conducted in a set of challenging environments.

### 4.1 Environment Settings

We used AntMaze, AntGather, and AntMaze-Sparse environments from the GYM MuJoCo library (Todorov et al., 2012). The first two involve manipulating a multi-armed robot to perform a set

of real-world tasks, while AntMaze-Sparse is the sparse reward version of AntMaze, presenting a greater challenge as it only provides a reward upon reaching the goal. For the state representation dimension, we selected a subset of raw state dimensions that contains spatial information (e.g. coordinates and arm angles) to serve as the node representation for the graph encoder-decoder.

## 4.2 EXPERIMENTAL COMPARISONS

We incorporated G4RL in the following existing GCHRL methods:

- **HIRO** (Nachum et al., 2018b): This is the first method which describes how the Goal-conditioned information can be integrated into hierarchical agents.
- **HRAC** (Zhang et al., 2022): This method enhances the performance of HIRO by training an adjacency network that produces subgoals easier to reach from the current subgoal.

In addition to comparing these two GCHRL-G4RL methods with their native counterparts, we also compared them with the following non-hierarchical method:

- **TD3** (Fujimoto et al., 2018): This is a well-known non-hierarchical policy-based method designed for continuous action spaces and we use it to implement both high- and low-level agents.

Although reward is a key metric of an agent's learning ability, for AntMaze and AntMaze-Sparse, we compare success rates of these methods instead of their rewards on the corresponding tasks. This is because higher rewards in AntMaze/AntMaze-Sparse do not necessarily indicate better performance; the agent may still achieve high rewards without reaching the goal.

The learning curves of baseline methods and G4RL-applied versions are plotted in Figure 1. Note that all the curves reported in Section 4 are averages from 5 independent runs and they have been smoothed equally for better visualization.

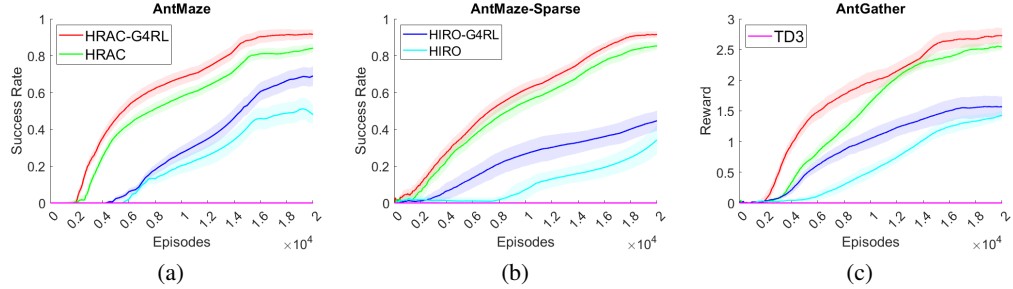

Figure 1: Success Rate on (a) AntMaze (b) AntMaze-Sparse and Reward on (c) AntGather, using HIRO, HIRO-G4RL, HRAC, HRAC-G4RL, and TD3. The legends are distributed across sub-figures for better visualization.

From Figure 1, we observe that in all three environments, incorporating G4RL in HIRO and HRAC significantly enhances their performance, further improving the already strong results of these hierarchical methods compared to the non-hierarchical method.

## 4.3 ABLATION STUDY

This section discusses the different variants of G4RL to show the effectiveness of adding high-level and low-level intrinsic rewards. The following variants are compared in this part:

- **High+Low-level intrinsics**: Apply both equation (9) and equation (10) to the high-level and low-level rewards respectively.
- **High-level intrinsic only**: Apply equation (9) to the high-level rewards and set $\alpha_l = 0$ in equation (10) when it is applied to the low-level rewards.

- **Low-level intrinsic only**: Apply equation (10) to the low-level rewards and set $\alpha_h = 0$ in equation (9) when it is applied to the high-level rewards.
- **HIRO/HRAC**: Vanilla baseline methods.

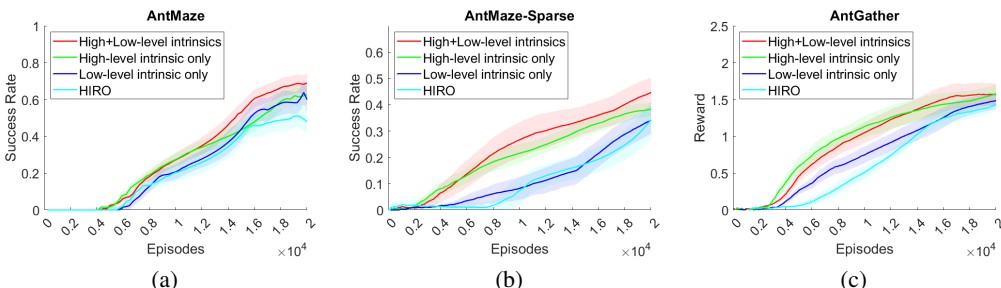

Figure 2: Success Rate on (a) AntMaze (b) AntMaze-Sparse and Reward on (c) AntGather using HIRO-G4RL, HIRO + High-level intrinsic, HIRO + Low-level intrinsic and HIRO.

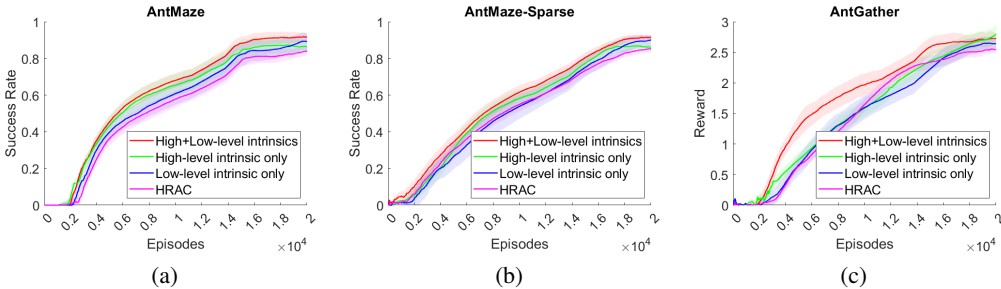

Figure 3: Success Rate on (a) AntMaze (b) AntMaze-Sparse and Reward on (c) AntGather using HRAC-G4RL, HRAC + High-level intrinsic, HRAC + Low-level intrinsic and HRAC.

Figures 2 and 3 show that for both HIRO and HRAC, G4RL with the intrinsic reward applied to both high-level and low-level agents achieves the best result across almost all three tasks. Meanwhile, the high-level intrinsic only variant tends to perform better than the low-level intrinsic only variant in our selected tasks, particularly in AntMaze-Sparse. This suggests that in sparse reward tasks, the high-level intrinsic reward proposed in equation (9) can facilitate efficient exploration.

### 4.4 SUBGOAL SPACE VISUALIZATION

This section shows how the subgoal space evolves in the AntMaze environment as the number of training episodes grows. We recorded state representations encountered in specific episodes and then used the corresponding graph encoders from those episodes to map these state representations to the subgoal representations. The subgoal representations are projected into 2D using PCA for visualization.

The distributions of subgoal representations in the subgoal space across different episodes are shown in Figure 4.

From the figure we can conclude that the graph encoder-decoder gives better subgoal representations when the number of episodes grows. At the 2000th episode, the subgoal representations form a few chains in the 2D space, indicating that the current graph is partitioned into many isolated sub-graphs. The graph encoder has not yet learned to understand the environment well, and the agent cannot find relationships between different state representations yet. By the 4000th episode, the representations are less isolated, suggesting that the graph encoder has begun to learn the connections between different groups of state representation. Finally, at the 6000th episode, the subgoal representations are distributed more evenly across the entire space, indicating that the encoder can organize them

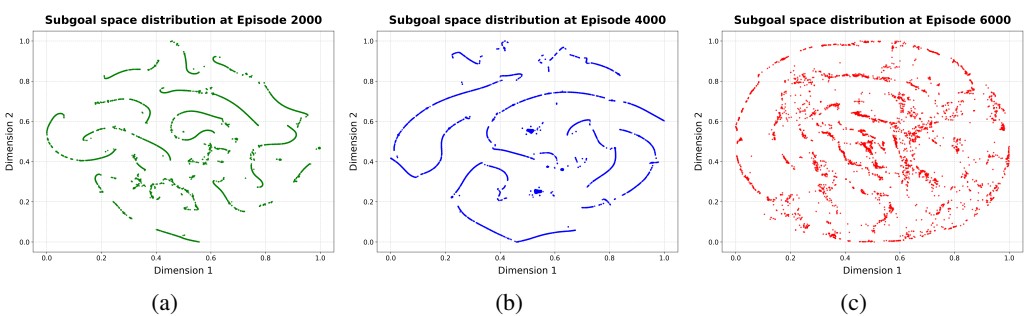

Figure 4: The distribution of subgoal representations in the subgoal space at the (a) 2000th (b) 4000th (c) 6000th episode.

into a highly connected graph. The subgoal space now better represents the underlying problem's structure.

The evolution of the graph encoder demonstrates that it can indeed learn to represent the task structure as the number of episodes grows.

## 5    CONCLUSION AND FUTURE WORK

We present a novel approach using a graph encoder-decoder to mitigate the issues of poor subgoal representations and sample inefficiency in GCHRL. The architecture is designed to efficiently evaluate unseen states by operating in the graph representation space. Our proposed architecture is easy to implement and can be integrated into any prevailing GCHRL algorithms for better performance. Additionally, we provide a comprehensive explanation of why the proposed architecture works, supported by experiments on both sparse and non-sparse control tasks that demonstrate the effectiveness and robustness of our method.

Generalizing knowledge learnt in one environment to other similar environments remains to be solved in reinforcement learning. In the future, we will extend our work by exploring how to generate subgoals with more interpretable representations for knowledge transfer with the help of other kinds of graph representations (e.g. Graph Laplacian). Another direction is to transfer the knowledge stored in the graph to other tasks by analyzing graph structures and mapping nodes of their state graphs.

## 6    REPRODUCIBILITY STATEMENT

We included the implementation details of the proposed architecture, the environmental settings, and the hyperparameter settings in the appendix.

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

## A    IMPLEMENTATION DETAILS

### A.1    ENVIRONMENT DETAILS

**AntMaze** This environment is a part of the Gymnasium-Robotics libraries. The size of the environment is $24 \times 24$. Both of the state space and the action space are continuous, with a state dimension of 31 and an action dimension of 8. The reward of each step is defined by its negative Euclidean distance from the current location to the target position. At evaluation time, the goal is set to $(0, 16)$ and an episode is recognized as successful if the agent is within an Euclidean distance of 5 from the goal.

**AntMaze Sparse** This environment is a variant of Antmaze. The size of the environment is $20 \times 20$. The state and action spaces are the same as those in AntMaze. The reward of each step is 1 if and only if the agent reaches within an Euclidean distance of 1 from the goal, which is set to $(2, 9)$.

**AntGather** This environment is described in Duan et al. (2016). The size of the environment is $20 \times 20$. The state and action spaces are continuous. The task involves gathering apples to the designated place. The agent will be awarded $+1$ for each apple gathered and $-1$ for each bomb gathered. Apples and bombs are randomly placed in the $20 \times 20$ world.

### A.2    NETWORK ARCHITECTURE DETAILS

Our network architecture for the HRL agents is the same as described in Nachum et al. (2018b) and in Zhang et al. (2022). For HIRO and HRAC, both the high-level and low-level agents use the TD3 algorithm. The size of each hidden layer in actor and critic networks in TD3 is 300.

For the graph encoder-decoder, we use a four-layer fully connected network with the hidden size of 128. The activation function used is ReLU. The decoder is a dot product of the two input subgoal representations.

We use Adam as the optimizer for the actor network, critic network and the graph encoder is Adam optimizer (Kingma & Ba, 2014).

## B HYPERPARAMETERS

In this section we list all hyperparameters used in our experiments.

| Hyperparameters | Values |
| --- | --- |
| High-level agent | |
| Actor learning rate | 0.0001 |
| Critic learning rate | 0.001 |
| Batch size | 128 |
| Discount factor $\gamma$ | 0.99 |
| Policy update frequency | 1 |
| High-level action frequency | 10 |
| Replay buffer size | 20000 |
| Exploration strategy | Gaussian($\sigma = 1$) |
| Low-level agent | |
| Actor learning rate | 0.0001 |
| Critic learning rate | 0.001 |
| Batch size | 128 |
| Discount factor $\gamma$ | 0.99 |
| Policy update frequency | 1 |
| Replay buffer size | 20000 |
| Exploration strategy | Gaussian($\sigma = 1$) |

Table 1: Hyperparameters used in high- and low-level TD3 agents.

| Hyperparameters | Values |
| --- | --- |
| Number of nodes $N$ | 200 |
| Batch size | 128 |
| Optimizer learning rate | 0.0001 |
| $\epsilon_d$ | 0.1 for AntMaze/0.2 for others |
| $\alpha_h$ | 0.1 |
| $\alpha_l$ | 0.1 |
| $\beta$ | 0.2 |

Table 2: Hyperparameters used in the graph encoder-decoder.

