# OpenReview forum: "Learning subgoal representations from state graphs in goal-conditioned hierarchical reinforcement learning"
_ICLR.cc/2025/Conference — ICLR 2025 Conference Withdrawn Submission_

### Official Review · Reviewer_j8kK · 2024-10-26

**Soundness:** 2
**Presentation:** 3
**Contribution:** 2
**Rating:** 5
**Confidence:** 4

**Summary:**

This paper proposes a novel architecture that employs a graph encoder-decoder to summarize spatial information into subgoal representations and constructs a world model based on the state graph for the agent to generate auxiliary rewards in both the high-level and low-level policies.

**Strengths:**

The paper introduces the G4RL approach with a degree of originality, and the presentation is clear, effectively explaining the proposed method in a way that is easy to follow.

**Weaknesses:**

- The primary experiments are conducted in a limited range of environments.
- The ablation studies are insufficient, lacking a comprehensive analysis of key parameters such as $\epsilon_{d}$, $\alpha_{h}$, $\alpha_{l}$, and $N$. The existing experimental results do not adequately support the significance of these parameters as stated in the methods section.
- There is no comparison with other representation methods to demonstrate the advantages or disadvantages.
- The learned world model is influenced by the current policy distribution, and it may not accurately reflect the actual world model.

**Questions:**

- How is the state representation function $\phi$ implemented? For example, is it based on neural networks or dimensionality reduction? Please provide specific details of the implementation.
- What impact do the values of $\epsilon_{d}$ and parameters $\alpha_{h}$, $\alpha_{l}$, and $N$ have on the algorithm's performance?
- Can it be qualitatively or quantitatively demonstrated that the graph encoder-decoder effectively extracts spatial information?
- Has the representation of subgoal spatial distance been compared with other methods, such as [1]? Does it show advantages over these approaches?

If the author can address the aforementioned weaknesses and questions, I will consider increasing the score.

[1]Park, Seohong, Tobias Kreiman, and Sergey Levine. "Foundation Policies with Hilbert Representations." Forty-first International Conference on Machine Learning.

---

### Official Review · Reviewer_1153 · 2024-11-02

**Soundness:** 1
**Presentation:** 2
**Contribution:** 1
**Rating:** 3
**Confidence:** 2

**Summary:**

This paper presents a novel graph encoder-decoder approach designed to evaluate previously unseen states, which can be integrated with existing GCHRL (Graph-based Hierarchical Reinforcement Learning) methods. The proposed model is trained on state graphs generated during exploration, and the authors demonstrate its effectiveness through empirical evaluation. Results indicate improved performance in both dense and sparse reward environments, driven by multi-level intrinsic rewards derived from the graph encoder-decoder.

**Strengths:**

* The paper is clearly written and easy to follow.
* The proposed method improves upon baseline methods in the AntMaze and AntGather tasks.

**Weaknesses:**

* The proposed method does not function as a subgoal representation learning approach but rather predicts state affinity.
* The paper lacks strong positioning within the subgoal representation learning literature. It cites only one relevant work and does not provide adequate motivation or comparison with existing methods in this area.
* The method (G4RL) shares significant similarities with HRAC, raising several concerns: 1. G4RL constructs graphs by hard-thresholding distances in state feature space, while HRAC uses K-step affinity along trajectories. As a result, G4RL is both feature- and hyperparameter-dependent, introducing limitations. 2. HRAC applies a contrastive loss to ensure that the learned subgoal space adheres to a K-step adjacency constraint while preventing subgoals from being too close. How does G4RL regularize representation learning in the latent space? 3. What is the rationale behind combining G4RL with HRAC (i.e., HRAC-G4RL)? Does G4RL require HRAC's regularization in the latent space?
* The evaluation is limited in several respects: 1. The method is only tested on the AntMaze and AntGather tasks. 2. It is only compared to two pre-2020 methods, HIRO and HRAC, without including more recent subgoal representation learning methods such as LESSON, HESS, and HLPS.
* There is insufficient analysis of the method's sensitivity to hyperparameters, such as how \epsilon depends on the environment and state space features.

**Questions:**

Please address my questions in the weakness section

---

### Official Review · Reviewer_ngDJ · 2024-11-04

**Soundness:** 3
**Presentation:** 2
**Contribution:** 2
**Rating:** 3
**Confidence:** 3

**Summary:**

The paper focuses on Goal-conditioned Hierarchical Reinforcement Learning (GCHRL) setting and introduces a graph encoder-decoder that can evaluate unseen states and enhance performance. This encoder-decoder can be trained on data generated during exploration, and by leveraging the high and low-level intrinsic rewards from the graph encoder-decoder improves performance.

**Strengths:**

1. The paper focuses on an important problem of integrating graphs with Goal-conditioned Hierarchical Reinforcement Learning and improving performance.
2. The work provides a good motivation for the research problem and its importance.

**Weaknesses:**

1. The paper can benefit from improving the writing and cleaning up the list of their contributions.
2. The set of environments / task settings is limited and it would be beneficial to add more tasks.
3. In some results, the methods are pretty similar. Running more seeds or increasing the difficulty of the experiments could be useful to pull the methods apart.

**Questions:**

1. The settings and environments considered in the experiments are relatively simple. How does the method scale up?
2. How sensitive is the method to the value of K : the number of timesteps used by the high-level policy to propose a goal? Is it same across different tasks?
3. How many seeds were used for the experiments and how were they chosen?

---

> ### Author Response · Authors · 2024-11-19
>
> Thank you very much for the time and energy you have spent reviewing this paper.
>
> Here are our answers to your concerns point-by-point:
>
> 1. The settings and environments considered in the experiments are relatively simple. How does the method scale up?
>
> The strategies we proposed in this paper essentially act as add-ons to existing methods, thus as soon as those baseline methods can be applied to more complex environments, one can choose to add G4RL for better performance. We understand your concerns regarding the limited scope of experiments. The complexity of these experiments is more than it seems to the eye. To understand this, we need to consider that the complexities of our environments are focused on reasoning challenges such as sparse reward and partial observability instead of learning from complex visual observations. The truth is the environments that we have employed are very difficult for traditional agents to solve.
>
> Nevertheless, adding more tasks and environmental settings is beneficial and we are working on enriching our experiment part.
>
> 2. How sensitive is the method to the value of K : the number of timesteps used by the high-level policy to propose a goal? Is it same across different tasks?
>
> In all of our experiments, we set K to 10. Comprehensively, a large K would increase the variance of the learning signal and make the high-level agent more difficult to converge, while a small K would prolong the temporal scale of the planning process and impair HRL methods' advantage over non-hierarchical ones. We used 10 in all of our experiments because our method reaches its best performance under this setting.
>
> 3.How many seeds were used for the experiments and how were they chosen?
>
> Due to limited computing resources 5 seeds were used for each experiment and the seeds were chosen randomly.
>
> Thank you very much for your questions and suggestions!

---

### Official Review · Reviewer_tyfX · 2024-11-04

**Soundness:** 2
**Presentation:** 2
**Contribution:** 2
**Rating:** 3
**Confidence:** 3

**Summary:**

This paper presents a novel approach—Graph-Guided sub-Goal representation Generation RL (G4RL)—aimed at addressing several key issues faced by existing Goal-conditioned Hierarchical Reinforcement Learning (GCHRL) methods, including sample inefficiency and poor subgoal representations. By introducing a graph encoder-decoder architecture, G4RL effectively leverages the state graph generated during exploration to enhance the performance of existing GCHRL methods. Empirical results demonstrate performance improvements in both dense and sparse reward environments.

**Strengths:**

1.	Innovation: The introduction of a graph encoder-decoder offers a novel perspective on GCHRL, facilitating the online construction of state graphs that yield more effective subgoal representations.
2.	Generalizability: G4RL can be integrated into any existing GCHRL algorithm, making it versatile and applicable across various contexts.

**Weaknesses:**

1.	Increased Complexity: Although the graph encoder-decoder adds new functionality, the added complexity does not yield a substantial performance improvement over existing HRAC methods. This raises concerns about implementation and debugging challenges without corresponding benefits.
2.	Insufficient Comparisons: The paper lacks comparisons with several recent GCHRL methods, which limits the assessment of the proposed approach's advancements and advantages over established techniques.

**Questions:**

1. Complexity Management: How do the authors plan to manage the increased complexity introduced by the graph encoder-decoder in practical applications? Are there any proposed strategies to simplify the implementation while retaining performance benefits?
2. Comparison Metrics: What specific metrics do the authors plan to use in future work to compare G4RL against recent GCHRL methods? Will they consider not only performance but also computational efficiency of integration?

---

> ### Author Response · Authors · 2024-11-18
>
> Thank you very much for the time and energy you have spent reviewing this paper.
>
> The problem and suggestions mentioned are quite relevant to our paper so we organized them into the following points for better communication.
>
> 1. Complexity issue: Introducing the graph encoder-decoder indeed increases the complexity of the architecture thus leading to increased training time. To reduce the extra cost, we propose to stop updating the graph and freeze the parameters in the graph encoder when our architecture reaches the intended performance in practical applications.
>
> Another direction for simplifying the implementation is to use estimations of distances instead of distances deducted from sampled trajectories in the original state space for the training of the graph encoder-decoder. We tried to use distance estimators in unreported experiments and noticed that they could reduce the training time and still boost the original method's performance, although not as much as the version using actual distance as reported in our paper.
>
> 2. We will consider using metrics most suitable to the underlying task. To compare with recent GCHRL methods, we intend to use the same environments and metrics in their corresponding papers. For practical applications, we will focus on the metric(s) that are most related to the task objective.
>
> Also, adding the training/running time of G4RL and the underlying GCHRL method as a metric is necessary.
>
> Thank you again for your detailed review of our paper! We are grateful for your many suggestions and hopefully, our replies could help you understand our work better.

---

### Note · Authors · 2025-10-10

I have read and agree with the venue's withdrawal policy on behalf of myself and my co-authors.

---

### Meta-Review · Area_Chair_Zv9u · 2024-12-21

**Metareview:**

This paper addresses the limitations of integrating graphs with GCHRL by introducing a graph encoder-decoder that effectively evaluates unseen states, thereby enhancing subgoal representation and addressing sample inefficiency. The proposed method G4RL can be incorporated into existing GCHRL frameworks to improve performance, utilizing a network trained on state graphs generated during exploration.
Despite these contributions, according to the reviewers' feedback, this paper still has to be improved on several fronts, including addressing the increased complexity of the graph encoder-decoder without substantial performance gains, insufficient comparisons with recent GCHRL methods, limited task settings, and a lack of clarity in its contributions, as well as concerns regarding its positioning within the subgoal representation literature and the evaluation of its hyperparameter sensitivity.

**Additional Comments On Reviewer Discussion:**

The authors did not respond to all the reviewers, and the further rebuttal discussion is not responsive.

---

### Decision · Program_Chairs · 2025-01-22

Reject